# Development and assessment of a sustainable PhD internship program supporting diverse biomedical career outcomes

Patrick Brandt[1]*, Dawayne Whittington[2], Kimberley D Wood[3], Christopher Holmquist[1,4], Ana T Nogueira[1], Christiann H Gaines[1], Patrick Brennwald[1,5], Rebekah L Layton[1]*

[1]Office of Graduate Education, University of North Carolina, Chapel Hill, United States; [2]Strategic Evaluation, Inc, Durham, United States; [3]Department of Community and Public Health, Mountain Area Health Education Center (MAHEC), Asheville, United States; [4]The University of Texas, Tyler, United States; [5]Cell Biology and Physiology, University of North Carolina, Chapel Hill, United States

*For correspondence:
pdbrandt@email.unc.edu (D);
rlayton@med.unc.edu (RLL)

## eLife assessment

This **important** study evaluates the outcomes of a single-institution pilot program designed to provide graduate students and postdoctoral fellows with internship opportunities in areas representing diverse career paths in the life sciences. The data **convincingly** show the benefit of internships to students and postdocs, their research advisors, and potential employers, without adverse impacts on scientific productivity. This work will be of interest to multiple stakeholders in graduate and postgraduate life sciences education and should stimulate further research into how such programs can best be broadly implemented.

**Abstract** A doctoral-level internship program was developed at the University of North Carolina at Chapel Hill with the intent to create customizable experiential learning opportunities for biomedical trainees to support career exploration, preparation, and transition into their postgraduate professional roles. We report the outcomes of this program over a 5-year period. During that 5-year period, 123 internships took place at over 70 partner sites, representing at least 20 academic, for-profit, and non-profit career paths in the life sciences. A major goal of the program was to enhance trainees' skill development and expertise in careers of interest. The benefits of the internship program for interns, host/employer, and supervisor/principal investigator were assessed using a mixed-methods approach, including surveys with closed- and open-ended responses as well as focus group interviews. Balancing stakeholder interests is key to creating a sustainable program with widespread support; hence, the level of support from internship hosts and faculty members were the key metrics analyzed throughout. We hypothesized that once a successful internship program was implemented, faculty culture might shift to be more accepting of internships; indeed, the data quantifying faculty attitudes support this. Furthermore, host motivation and performance expectations of interns were compared with results achieved, and this data revealed both expected and surprising benefits to hosts. Data suggests a myriad of benefits for each stakeholder group, and themes are cataloged and discussed. Program outcomes, evaluation data, policies, resources, and best practices developed through the implementation of this program are shared to provide resources that facilitate the creation of similar internship programs at other institutions. Program development was

initially spurred by National Institutes of Health pilot funding, thereafter, successfully transitioning from a grant-supported model, to an institutionally supported funding model to achieve long-term programmatic sustainability.

## Introduction

A decade ago, in response to changing trends in the scientific workforce, the National Institutes of Health (NIH) published the NIH Biomedical Workforce Report (*Health, 2012*). The report acknowledged that most PhD training lacked experiential career development opportunities (*Alberts et al., 2014*) and it made suggestions for changes to doctoral-level career training that would lead to more diverse career outcomes. Soon after the report was released, the NIH Office of the Director created a new grant mechanism known as Broadening Experiences in Scientific Training (BEST), which spurred the development of novel career and professional development initiatives at many universities including the University of North Carolina at Chapel Hill (UNC) (*Laura Daniel and Chalkley, 2020*; *Lenzi et al., 2020*).

At the same time, momentum was gathering for career outcomes tracking for PhD graduates and transparent dissemination of these outcomes (e.g., NIH BEST data collection requirements; *Alberts et al., 2014*; *Stayart et al., 2020*). Hence, the movement to enhance PhD career training opportunities and provide transparent doctoral career outcomes tracking paralleled the evolving needs of the scientific workforce, all of which highlighted the need for doctoral and postdoctoral training programs to expand training to match the careers their trainees were likely to hold.

All NIH BEST grantees implemented various forms of experiential learning (*Lenzi et al., 2020*; *Van Wart et al., 2020*). Experiential learning can constitute a powerful approach to building career knowledge and skills. Examples of experiential learning include low-dose, short-term job simulations or site visits completed in a single day (*Collins et al., 2020*); medium-term courses developed specifically for doctoral students to gain business skills (*Petrie et al., 2017*), or shadowing over short or extended time periods; and longer-term high-exposure experiences such as internships (*Van Wart et al., 2020*). Graduate-level internships have been shown to have promise and successful outcomes (*Chatterjee et al., 2019*), including significant increases in career confidence (*Schnoes et al., 2018*), but are challenging to implement for a variety of reasons.

Although concerns exist that internships could impact time to a degree, evidence to date suggests that this is not the case (*Schnoes et al., 2018*; *Brandt et al., 2021*). Nonetheless, time invested by the interns as well as staff time and institutional resources required to manage an internship program can be barriers to program implementation, success, and sustainability. Internship programs at the doctoral and postdoctoral levels must have support from the faculty to create accessibility for trainees to participate. Faculty attitudes toward expanding career pathways for life science trainees have become an area of recent interest (*Watts et al., 2019*), but there remains much to be further explored on this topic. We hypothesized that faculty attitude toward trainee participation in time-intensive trainee career development such as internships, may improve because of positive experiences with the program, for example, by seeing trainees who maintain productivity and successfully navigate their doctoral training requirements throughout an internship experience. Further, we posited that communicating data about career trends and workforce realities to research advisors, along with the career development resources opportunities available to their trainees, is an important step to gaining faculty buy-in for high-dose career development training such as internships. Empirical data are needed to identify effective ways to facilitate faculty attitude change toward acceptance and encouragement of the diverse career pathways PhD trainees pursue, and the current work takes a step toward empirically examining that question.

The Immersion Program to Advance Career Training (ImPACT) internship program at UNC, which is the subject of the current research, was designed to allow experiential skill development in diverse career pathways tailored to the interests of trainees. For example, internships are available in the areas of pharmaceutical research and development, entrepreneurial grant writing, regulatory affairs, medical and regulatory writing, project management (e.g., contract research originations), medical affairs (e.g., medical science liaisons), science writing, science outreach, and college teaching to name a few. Internships are 160 hr long and can either be full-time for 4 weeks or part-time over 2–3 months. Interns require written support from their research advisor and must complete all qualifying exams

before applying. Usually, internships take place in the last 1–2 years before graduation. The internship application period opens in December with a showcase of posters presented by the prior year's interns. After a February deadline, interns are selected by program leaders based on the quality of the application, previous career exploration undertaken, ease of matching with a company partner, and diversity of internship interest. A program director meets with each internally selected intern to better understand their timeline, interests, and needs. The intern is primarily responsible for doing the groundwork to be placed but is supported and assisted by program directors in exploring, making connections, and deciding on the best internship and host match for them. In its simplest form, this may involve interviewing for an internship slot that was advertised by the host organization during the application period. If the intern is interested in a more tailored opportunity, then they can network with a potential host to create an internship. The host makes the final decision on whether to offer or accept an intern, and the intern can select if they have more than one offer. However, in most cases a small pool of possibilities is winnowed down to a 1:1 match, and only if that match is ultimately not selected do the interns/hosts move onto explore new options. Internships often take place in the summer but can occur any time of the year (e.g., teaching internships commonly occur over the fall semester).

The internship program has enjoyed remarkable success since its start in 2015 and provides a case study for the examination of the unique intersections between experiential learning, career development, program evaluation, stakeholder buy-in, and career outcomes tracking. It also permits the comparison of interns with non-intern controls to better assess the benefits of this and similar programs.

## Results

A rigorous program evaluation plan consisting of internal (program staff) and external (Strategic Evaluations, Inc) evaluation components was built into ImPACT from the beginning and allowed us to answer the following research questions using a mixed-methods approach (cross-stakeholder surveys and focus groups): (1) What are the benefits of the internship to each stakeholder group (interns, research advisors, and internship hosts)? (2) What long-term career outcomes are positively impacted by the internship experience? (3) Do faculty attitudes shift in a positive direction during the establishment of an effective internship program? We also captured lessons learned about program development annually to support formative changes, as well as to inform summative best practices. Results presented include graphical representations of quantitative survey results. The response rate for the 123 survey invitations sent to interns and their current research advisors and internship hosts ranged from 61% for research advisors to 73% for hosts, and about 66% for interns (averaging pre- and post-survey responses). In addition to quantitative surveys, qualitative themes and exemplars were collected from focus groups (see 'Methods' for details).

### Part 1: Internship benefits

#### Benefits to intern

By design, the main beneficiaries of the internships are the interns themselves. The intended impact was for interns to gain valuable skills, build a professional network of colleagues and mentors, learn how scientific and technical development occur in non-academic sectors, and get to prove themselves as valuable employees; indeed, the data support this (see *Figure 1*). To assess internship benefits, interns received pre- and post-surveys, and research advisors and hosts received post-surveys only (see 'Methods').

#### Well-received/well-implemented

Overall ratings from interns show that 90% were highly satisfied with their internship (*Figure 1e*). Ninety percent of interns also rated the support they received from their supervisor as good or very good, with less than 2% rating support as poor (*Figure 1a*). Ninety-seven percent of interns agreed that the internship provided networking and professional development opportunities that they would not have received otherwise (*Figure 1c*), and nearly 80% reported that they received coaching specific to their career path (*Figure 1b*).

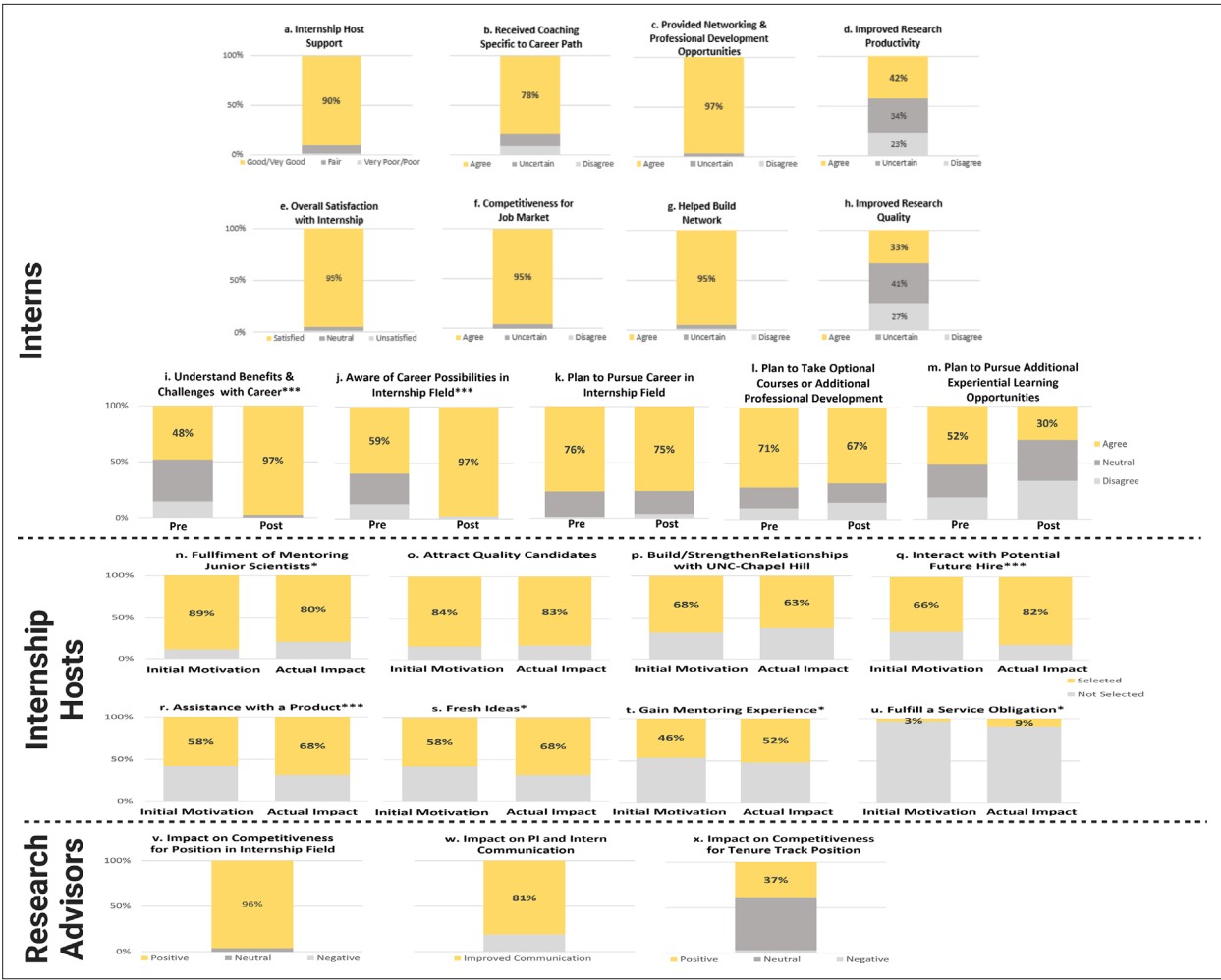

**Figure 1.** Benefits of internship for different stakeholders (quantitative evidence). (**a–h**) Benefits to interns documented through post surveys. (**i–m**) Benefits to interns documented through pre- and post-surveys. (**n–u**) Benefits to internship hosts documented through post surveys. (**v–x**) Benefits to research advisors documented through post surveys. Mean values for the benefits to the interns documented through pre- and post- (**i–m**) were tested for significance using an independent samples *t*-test. Asterisk(s) indicate(s) differences were statistically significant (*p<0.05; **p<0.01; ***p<.001).

## Benefits

Nearly all (95%) of interns agreed that the internship positively impacted their competitiveness for the job market (*Figure 1f*) as well as helped them build a network that they would rely upon for their career advancement (*Figure 1g*). Interestingly, 42% of interns agreed that the internship improved their research productivity, and another 34% said that it neither improved nor decreased their productivity (*Figure 1d*). Lastly, 33% of interns agreed that the internships improved the quality of their research and an additional 41% stated that it neither increased nor decreased the quality of their research (*Figure 1h*).

## Impacts (pre and post)

Survey data indicate that internships increased interns' knowledge of the career area they were exploring. The percentage of interns agreeing that they understood the benefits and challenges associated with a career in the field in which their internship focused *doubled* from pre- to post-surveys (*Figure 1i*), with only 48% of interns agreeing with this item on the pre-survey (mean = 3.4) versus 97% on the post (mean = 4.3). A similar increase was documented from pre- to post- as interns rated their awareness of career possibilities (*Figure 1j*) in the field of the internship. Only 59% of interns agreed with this item on the pre-survey (mean = 3.5) versus 97% on the post (mean = 4.2). Both areas,

*understanding benefits and challenges with career* and *awareness of possibilities in internship field,* showed statistically significant increases from pre- to post-internship (p<0.001).

Interns' plans to take optional courses and participate in additional professional development (*Figure 1l*) or experiential learning opportunities in areas related to the internship (*Figure 1m*) trended downward from pre to post. The downward trend suggests that the internships fulfilled their immediate desires to know more about the field in which their internship focused.

Nearly all (96%) of research advisors indicated that the internship had a positive impact on interns' competitiveness for a position in the field that was the focus of the internship (*Figure 1v*). Thirty-seven percent of research advisors also thought that the internship positively impacted interns' competitiveness for a tenure-track position (*Figure 1x*).

The external evaluation team conducted focus group interviews with research advisors at two time-points, Year 1 and Year 5 of the implementation. A group of faculty advisors in a range of disciplines and demographics, all of whom were active mentors with extensive training experience were invited to participate in the focus groups. Seven faculty advisors participated in the Year 1 focus group and five of those same seven participated in Year 5. Saturation can occur with as little as six interviews in homogeneous samples (*Guest et al., 2006*) such as our biomedical faculty research advisors at a single institution. (More details on the design of these focus group sessions are included in the 'Methods' section.) In the focus group with faculty that was conducted at the first timepoint (Year 1), several research advisors indicated that experiencing different careers would be helpful to trainees who may decide on non-academic career paths, especially given the challenging job market. Furthermore,

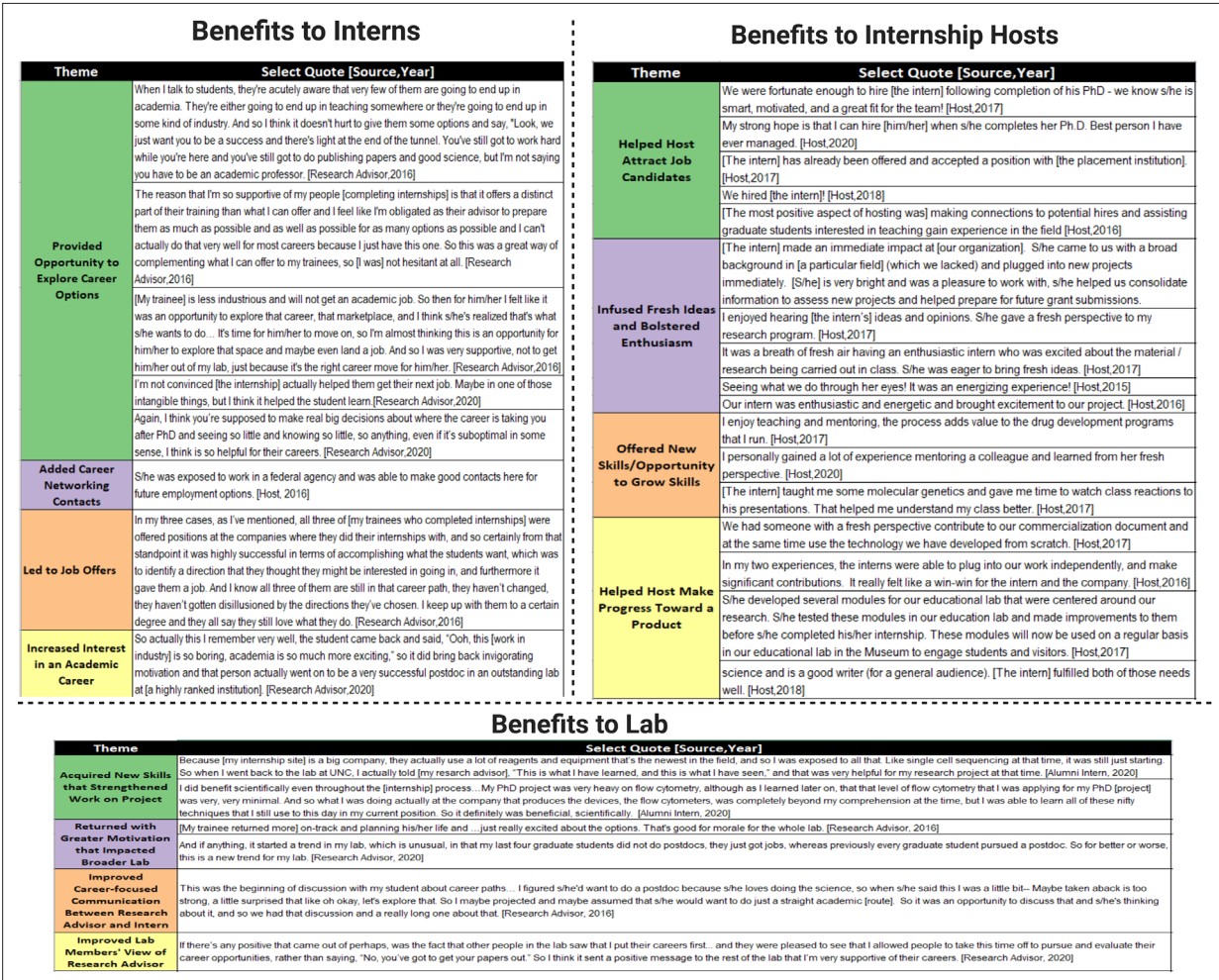

**Figure 2.** Benefits of internship for different stakeholders (qualitative evidence). Stakeholder interviews were moderated by an external evaluation team, with participants' identity remaining confidential. Interviews were transcribed and imported into Atlas.ti, with the evaluator developing codes for key themes. Representative quotes supporting each major theme that emerged are presented in the figure.

research advisors noted that even those trainees considering an academic career path might find internships to boost their CVs. Finally, internships were seen as an effective strategy for trainees to gain experience in areas outside their research advisor's expertise. The consensus was that internships would most benefit strong students who were likely to maintain focus on their research regardless of circumstances, or those trainees who seem more interested in a non-academic career (see *Figure 2* for supportive quotes). The inclusion of a second timepoint allowed faculty to reflect on the process after a subset of interns were able to complete the program, graduate, and transition to first positions after pre- and postdoctoral training. Comments from the second timepoint (Year 5) focus group suggest that trainees who accepted positions in the area of their internship remained satisfied with their chosen careers. Other faculty comments suggest that the internships helped trainees hone their career interests, even if they did not receive a job offer or accept a position as a result of the internship (see *Figure 2* for supportive quotes).

## Benefits to lab

The benefits of the internships extended beyond the interns themselves. Survey data from all the research advisors and interns also identified positive impacts in the lab. For instance, data suggested that the interns' labs also benefited when trainees implemented specific techniques learned during their internship. Moreover, in some cases the interns had renewed enthusiasm for their work after seeing other career paths, and their excitement enhanced the overall culture of the lab. Finally, internships provided opportunities to foster collaborations between the research lab and industry.

Roughly 80% of research advisors noted improved communication between themselves and their trainee who participated in an internship (*Figure 1w*). For example, one research advisor described how the program sparked an in-depth conversation with their mentee about career options and led them to question the assumptions they had made about their trainees. In interviews, research advisors often discussed how the improved communication led to more focused career conversations with their trainees. This improvement was attributed more to the structure of the broader internship program, particularly the inclusion of career interviews (see *Figure 2* for supportive quotes).

Research advisors also indicated that their trainees' exposure to different work environments not only helped them clarify career paths but also allowed them to pass information to other trainees in the lab about various organizational cultures. A subset of research advisors witnessed enhanced motivation among their trainees toward a chosen career path and/or increased interest in an academic career. Interview data indicate that interns in some cases brought their ideas about the value of their profession back to other trainees, which tended to have the effect of boosting motivation throughout the lab. Lastly, research advisors noticed that allowing trainees to participate in the internship program demonstrated to others in their lab that trainees are valued. The image of the research advisor was improved when others saw that trainees' career priorities were taken into consideration (see *Figure 2* for supportive quotes).

Alumni focus group interviews (see 'Methods' for details) revealed that a subset of interns returned to the lab with a better understanding of techniques and new instrumentation. In some cases, these new skills strengthened their work on their project, while in other cases the skills served them well as they transitioned into their career positions.

## Benefits to internship host

An analysis of internship hosts' initial motivations to host an intern shows that most of them participated for two primary reasons: (1) to enjoy the fulfillment of mentoring a junior scientist (89%) (*Figures 1n and 2*) to attract a quality applicant to their field (84%) (*Figure 1o*). Roughly two-thirds of supervisors were also motivated to host an intern to build a relationship with UNC as well as interact with a potential future hire (*Figure 1p and q*). Lowest on the list in terms of motivations for internship hosts to host an intern was the fulfillment of a service obligation (i.e., they were asked to do so by a supervisor, *Figure 1u*).

Furthermore, internship hosts were asked to rate the extent to which their initial motivations to host an intern aligned with the actual impacts they experienced in hosting an intern. Ratings show that more than 80% of supervisors indicated that hosting an intern highly impacted three areas: (1) fulfillment by mentoring (89%) (*Figure 1n*), (2) attraction of a quality applicant to their field (83%) (*Figure 1o*), and (3) interaction with a potential future hire (82%) (*Figure 1q*). Roughly two-thirds of supervisors

**Table 1.** Logistic regression model of internship participation, number of career interests, and trainee type (graduate student versus postdoc) on identical interest match - (Model 1).

| Variables | p-Value | OR | 95% CI Lower | Upper |
|---|---|---|---|---|
| Internship participation | <0.001 | 2.99 | 1.48 | 6.06 |
| Number of career interests | <0.001 | 1.20 | 1.11 | 1.29 |
| Trainee type (postdoc) | <0.001 | 5.82 | 3.76 | 9.01 |

indicated high impacts in three additional areas: building a relationship with UNC (*Figure 1p*), gaining assistance with a product and/or grant (*Figure 1r*), and gaining fresh ideas (*Figure 1s*).

Interacting with a potential hire (p<0.001) increased significantly, which indicates that the internships were even more valuable than expected in accomplishing one of the primary goals of the program. In addition, interns were even more valuable than expected by hosts in *assisting with a product* (p<0.001) and *providing fresh ideas* (p<0.05). Lastly, a largely unanticipated benefit which also significantly increased pre- to post-internship hosting experience was *fulfilling a service obligation* (p<0.05), which perhaps came with recognition post participation from their organization. Interestingly, while gaining fulfillment in mentoring a junior scientist was initially high and maintained high levels, it still showed a significant decrease from pre- to post-internship, suggesting that the actual hosting of an intern may have been less fulfilling than hosts initially anticipated.

## Part 2: Career outcomes

We expected that a primary benefit of completing an internship would be that interns are more likely, compared to controls, to find a match between their desired career path and their first position post-training period (e.g., after graduate/postgraduate program or position). Trainees who had completed either (1) an entrance survey documenting their career interests or (2) a pre-internship career interest survey, and who had transitioned to their first-destination job, were included in this analysis. This allowed for a comparison of first-destination career outcomes for trainees who completed an internship with controls who did not do an internship. NIH BEST entrance survey data and pre-internship survey data were used to assess career interests, and first job placement title and employer were gathered using publicly available information (i.e., LinkedIn profiles, see 'Methods' for details; career classifications based on the UCOT, *Stayart et al., 2020*). This analysis compared matches between interns and control trainees who did not complete an internship (non-interns). A match was defined as the overlap between the field of first position post-training and the field of career interest as reported in the pre-internship survey or the NIH entrance survey.

Logistic regression was used to test if participation in the internship significantly impacted career interest-first job placement match. Two variations were tested: an identical match or a group interest (see 'Methods' for details). Interestingly, the model was statistically significant for both identical and group matches, with nearly identical patterns and values in each scenario. Therefore, we report below the more stringent identical career match.

The overall model was statistically significant ($\chi^2$ = 92.39, $R^2_{Nagelkerke}$ = 0.195, p<0.001). While controlling for the number of career interests and trainee type, we found that internship participation

**Table 2.** Matches between career interest and first job placement.

| | n | Exact match (%) |
|---|---|---|
| **All interns** | **130** | **51 (39%)** |
| Graduate students | 109 | 42 (39%) |
| Postdocs | 21 | 9 (43%) |
| Non-interns | 578 | 165 (29%) |
| Graduate students | 251 | 57 (23%) |
| Postdocs | 327 | 108 (33%) |

**Table 3.** Summary for regression model controlling for demographic variables race/ethnicity (UR/WR), gender (female/male), and citizenship (citizen/international) (Model 2).

| Variables | p-Value | OR | 95% CI | |
|---|---|---|---|---|
| | | | Lower | Upper |
| Internship participation | <0.001 | 3.51 | 1.71 | 7.21 |
| Number of career interests | <0.001 | 1.20 | 1.11 | 1.29 |
| Trainee type (Pd) | <0.001 | 5.03 | 2.94 | 8.60 |
| Gender* | 0.02 | 1.58 | 1.07 | 2.33 |
| Race/ethnicity (ns) | 0.52 | 1.00 | 1.00 | 1.01 |
| Citizenship (ns) | 0.26 | 1.00 | 0.99 | 1.00 |

significantly predicted an identical match between an intern's first job placement and their career interests as defined before the internship. The odds ratio for a match after doing an internship was 2.99, indicating that interns were nearly three times more likely to match with their career interests than non-interns (p<0.001, *Table 1*). Unsurprisingly, for interns and controls, more career interests were associated with an increased chance of a match (p<0.001, OR = 1.20). Of note, postdocs, whether they did an internship or not, were nearly six times more likely to match their career interest than graduate students (p<0.001, OR = 5.82). This is likely due to the proximity of the career interest survey response to when the postdoc finds themself on the job market. It also is indicative of postdocs, in general, having better-defined career interests compared to graduate students. Because of the variable timing between pre-internship career interest surveys among interns and control trainees and securing the first job, future studies could more rigorously evaluate changes in career preferences between pre and post internship with an analysis that considers the time that has elapsed between career interest noted pre-internship versus post-internship career placement.

When these same data are examined as percentages of trainees whose first position matches their top career interest, we find that 51 out of 130 (39%) interns had an exact match, whereas only 165 out of 578 (29%) non-interns had an exact match (see *Table 2*).

Furthermore, post hoc tests explored the potential impact of social identity groups on career matching, but trends were nearly identical, hence the original, simple Model 1 was retained. Model 2 included demographic information (race/ethnicity [UR/WR], gender [female/male], and citizenship [citizen/international]). Race/ethnicity and citizenship did not impact the model, indicating that UR and international applicants were as likely as WR and citizen applicants to match their career interests (p<0.52, OR = 1.00 and p<0.26, OR = 1.00, respectively). For this reason, we ran a simplified model including only gender as a demographic variable. The trends were similar for career interests and trainee type (*Table 3*). The overall model remained statistically significant while controlling for gender, trainee type, and career interests, with nearly identical patterns and significance levels for the impact participation (OR = 3.51, p<0.001) ($\chi^2$ = 97.57, $R^2_{Nagelkerke}$ = 0.21, p<0.001). Interestingly, applicants who identified as female were nearly one and a half times more likely to match their career interests when compared with applicants who identified as male (p=0.02, OR = 1.58).

One goal of the internship program is to decrease the percentage of graduate students who pursue postdoctoral training when such training is not necessary. Postdoctoral training is an excellent and required training for some research-intensive career paths, namely academic tenure track positions. However, some graduate students enter postdoctoral training by default because they are unsure of what profession they plan to pursue, and they see a postdoc as the best way to keep 'all their doors open'. Not all career paths require postdoctoral training, some hiring managers in the industry view a lengthy postdoc training period negatively when considering candidates, and postdoctoral training

**Table 4.** Rates of postdoctoral training for interns and non-interns.

| Graduate student participant status | n | Postdoc rate (%) |
|---|---|---|
| Interns (participants) | 107 | 36 |
| Controls (non-participants) | 499 | 57 |

**Table 5.** Job sector and career type of first job after graduate student internships.

| Tier | Interns | Non-interns |
|---|---|---|
| Job sector | Total n=107 | Total n=499 |
| Academia | 30 (28%) | 271 (54%) |
| For-profit | 67 (63%) | 156 (31%) |
| Government | 2 (2%) | 41 (8%) |
| Non-profit | 5 (5%) | 30 (6%) |
| Unknown | 3 (3%) | 1 (<1%) |
| Career type | | |
| Primarily research | 65 (61%) | 381 (76%) |
| Primarily teaching | 6 (6%) | 19 (4%) |
| Science related | 32 (30%) | 89 (18%) |
| Not related to science | 0 (%) | 1 (<1%) |
| Other/unknown | 4 (4%) | 9 (2%) |

has been shown to decrease lifetime earnings and delay retirement savings (*Kahn and Ginther, 2017*). We analyzed whether interns were more likely to enter the biomedical workforce directly instead of doing a postdoc when compared to non-interns and found that 38 of 107 interns (36%) who had graduated between 2015 and 2021 continued their training in a postdoctoral position. Looking at non-intern controls who graduated during that same time span, we found that of 499 graduation date-matched non-interns, 283 (57%) pursued postdoctoral training (*Table 4*).

Without a doubt, trainees who elect to do an internship are a self-selected population with career biases. Interns are less likely to want to pursue an academic tenure track position, they are more likely to have for-profit career aspirations, and, in the case of our program at least, they must have the support of the research advisor for doing the internship. Despite these inherent biases, it is valuable to examine the different career outcomes of interns versus non-interns (*Table 5*). For this analysis, only graduate student interns and graduate student controls (matched by graduation year) were examined. Nineteen percent of interns were employed in the academic sector as their first position post-graduation compared to 46% for non-intern controls. In contrast, 75% of interns were employed in the for-profit sector compared to 42% of non-interns. When examining career type, the numbers were not as disparate; 57% of interns took primarily research positions defined as positions where they are generating or analyzing scientific data. Non-interns took primarily research positions at a slightly higher rate – 68%. Interns were more likely to take positions in science-related careers (37%) compared to non-interns (24%).

## Publication profiles of interns and non-interns

One concern cited by faculty reluctant to support a trainee's internship is the potential for reduced productivity on the part of the intern. Results from a previous study across 10 institutions (*Brandt et al., 2021*) show no delay in graduate training nor a reduction in research productivity (first author or total publications), for trainees participating in professional development activities including internships. We tested this hypothesis again on local programmatic data in an expanded group with a larger number of both interns and controls. To give graduates time to complete publications with their research advisor, interns and controls were only included in this analysis if they had graduated 2 years

**Table 6.** Type and quantity of publications between graduate student interns and non-interns.

| Graduate student participant status | n | **Average** first-author publications (p=0.52) | **Average** total publications (p=0.66) |
|---|---|---|---|
| Interns (participants) | 54 | 1.94 | 3.65 |
| Controls (non-participants) | 410 | 1.81 | 3.85 |

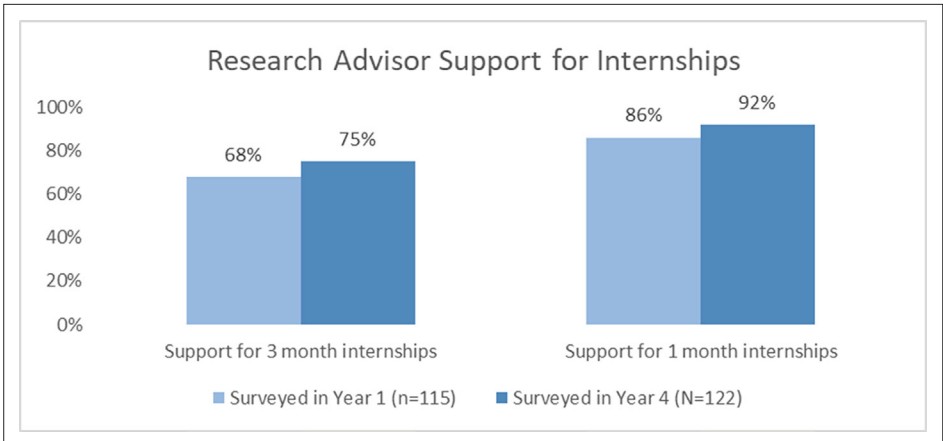

**Figure 3.** Faculty ratings for research advisor-support of 1- and 3-month internships.

or more prior to the analysis date. In this larger study, 54 interns have, on average, slightly more first-author publications and slightly fewer total publications but neither difference is statistically significant (*Table 6*).

We also examined average time to degree (defined as the start date to date of degree conferral for 107 graduate student interns and 420 non-interns, date-matched controls). The average time to degree conferral for interns was 67.1 months and for controls was 69.3 months (independent samples *t*-test, p=0.09). Therefore, as in our previous work (*Brandt et al., 2021*), we find no evidence of any deleterious effect on graduate trainee productivity as a consequence of participating in a 1-month internship program.

## Part 3: Changes in faculty attitude toward internships

Before the internship program began (Year 0), we surveyed active training faculty to ask them how likely they would be to support one of their students in good standing if the student asked permission to participate in an externally paid 1- or 3-month internship. Five years later we asked the same group of faculty the same question. Independent samples *t*-tests were used to compare faculty attitudes toward both lengths of internship at Year 0 and Year 5. Significant improvements in faculty support were observed for both durations of internship participation (3-month internship, $t(229)=2.54$, p=0.01; 1-month internship, $t(212)=2.00$, p<0.01); unequal variances accounted for as appropriate. Across the 3- and 1-month options, responses of both *highly unlikely* and *unlikely* decreased, and a higher portion of those in the *likely* category moved to *highly likely* (*Figure 3*). It is noteworthy that support for 1-month internships grew to 92% by Year 5 of the program.

## Part 4: Lessons learned and advice for emerging internship programs
### Length and structure of the internship
UNC's ImPACT internship program was started in 2015 and at the writing of this article has placed 175 interns with 80 partnering organizations. The program is well-known and appreciated among trainees and research advisors, and it is a major recruiting tool for UNC's life science PhD programs. Many of the lessons we have learned from the program's success may be helpful to others who are developing their own life science internship programs for PhD trainees and postdocs. The structure of the internship experience is integral to its success as a win-win experience for the three major stakeholders – interns, research advisors, and internship hosts. Research advisors, as stated above, are widely supportive of 1-month internships and most consider any internship longer than 3 months a non-starter. Companies, on the other hand, are used to 3–6-month-long internships, especially when they are paying for the intern. Interns' preferences for length of experience typically lie somewhere in between. They want to stay on track in their own research projects, but they also want to get as much skill development and networking as possible in the internship.

Because the institution pays for the interns' time during the internship program, we were able to settle on 160 hr internships as the standard. These can take place full-time over 4 weeks or part-time

over 2–3 months. A 1-month, full-time internship is the preferred model of most research advisors. Research advisors in both focus group interviews noted that this format is simpler than a longer, part-time internship because it is least disruptive to trainees' progress in the lab and requires less buy-in from stakeholders. This finding held true no matter which cohort was interviewed or how many interns the research advisor had mentored. Some research advisors indicated that trainees in many cases take month-long leaves for other reasons, and advisors often did not even notice the trainees' absence when the internships were brief. One noted that whether or not the trainee was working in the lab for 4 weeks was 'irrelevant to me', and another stated that they would not have objected to continuing to pay for the trainee during the internship month. However, our survey's results also suggested that limiting the time of the internship was critical for overall faculty advisor buy-in. Support for the internship consistently dropped as the length of the internship proposed increased from 1 month to 3 months (*Figure 3*). One advisor stated their opinion that semester-long internships 'would completely derail' trainees.

Trainees' abilities and skills are also factors that help determine the success of an internship. Although scientific capacity is certainly important, research advisors also considered trainees' professionalism and time management when determining whether an internship would be appropriate and what model would work best. For example, highly organized students could benefit from longer, part-time internships that allowed them to spend time in the lab, but they felt that this model would not work for those who were unable to maintain the balance of lab and internship responsibilities. Trainees' communication skills and level of focus were also deemed important to consider.

The timing of the internship within the trainee's career was also essential. Internships were seen as least disruptive when trainees were close to graduation, or training completion in the case of post-docs. Moreover, it was especially helpful for trainees to be in writing rather than conducting research because their presence in the lab was not as critical. The technical expertise and scientific knowledge gained as a PhD student nears graduation are valuable to the internship host and they also prefer interns who are more senior, especially given the relatively short duration of the ImPACT internships.

Important design features cited by multiple stakeholders include pre-internship career coaching, a poster session highlighting the work of the interns, and the scope of work documents (each described in more detail below). Pre-internship career coaching is critical so that the intern approaches the internship as a skill-building experience, not simply a career exploration experience. One of the review criteria for intern applications is whether they have taken advantage of career development workshops, networking events, and other resources and how these experiences point them to the internship they are interested in.

A poster session highlighting aspects of each internship project (non-confidential portions) is held at the North Carolina Biotechnology Center (state-wide economic development organization) at the end of each year. Internship supervisors and research advisors are invited to attend as are graduate students who plan to apply for an internship in the coming year. The event has aspects of a typical scientific poster session, a networking event, and a program information session – the event is an important and popular part of the yearly cycle of the program.

It is important that all stakeholders agree to the scope and deliverables of the internship. To facilitate this, a written agreement detailing the expectations of the internship experience is collaboratively drafted and signed by the intern, the internship host, and the research advisor. The so-called Scope of Work also contains the names and contact details of the parties, the start and end dates of the internship, a description of the project, the deliverables expected at the end of the internship, and other details as applicable. Common deliverables include a written report submitted by the intern to the host or a slide presentation by the intern given to the hosting unit. In some cases, the entire internship project is dedicated to the deliverable such as writing an NIH Small Business Research Innovation grant for a startup company or creating a competitor analysis report for an emerging product. The scope of work is signed by the intern, the research advisor, and the internship supervisor.

## Stakeholder engagement and satisfaction

Ensuring a positive experience for all stakeholders is critical to the success and sustainability of an internship program. *Figure 4* shows the results of surveys completed by internship hosts and research advisors and provides insight into the challenges of meeting all stakeholder needs. Overall, 95% of internship hosts were satisfied with the internship program (*Figure 4h*), with 96% being highly likely

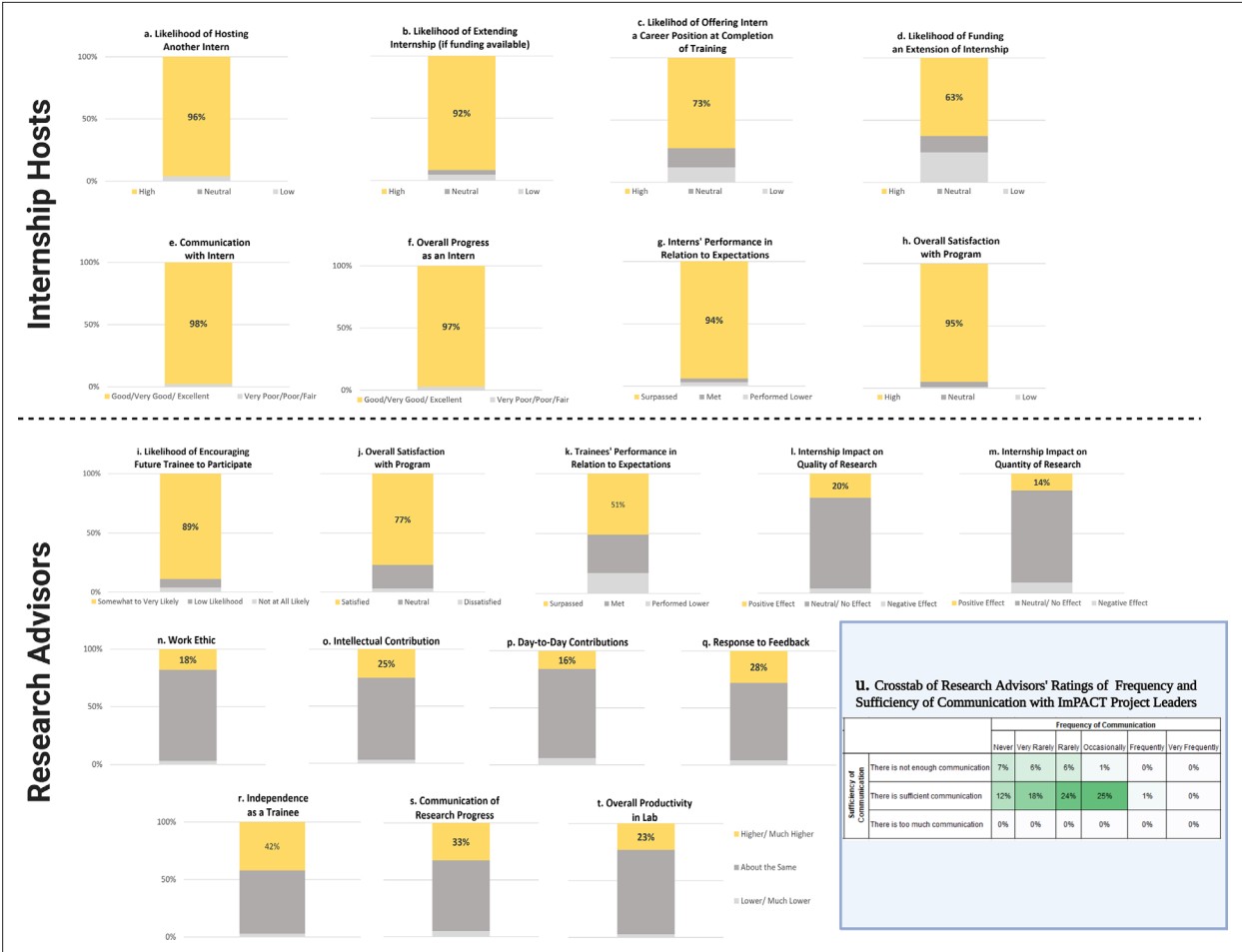

**Figure 4.** Sustainability and lessons learned (quantitative evidence). (**a–h**) Internship host ratings of various internship components that are proxies for sustainability. (**i–t**) Research advisor ratings of various internship components that are proxies for sustainability. (**u**) A crosstab of research advisors' ratings of frequency and sufficiency of communicating with project leadership.

to supervise an intern in the future (**Figure 4a**). Nearly all (92%) of internship hosts indicated that they were highly likely to extend the internship if funding was made available (**Figure 4b**) and 63% indicated they were highly likely to extend the internship even if they had to provide the funding (**Figure 4d**). Overall, 98% of internship hosts rated communication with their intern as good, very good, or excellent (**Figure 4e**), and 97% of hosts rated their intern's overall progress similarly positive (**Figure 4f**). Satisfaction with the intern was extremely high, with 94% surpassing hosts' expectations (**Figure 4g**). Nearly three-quarters (73%) were highly likely to offer their intern a career position within their organization after completion of UNC training (**Figure 4c**).

Nearly 80% of research advisors indicated they were satisfied with the program (**Figure 4j**). Trainees' skills across several areas were rated as about the same or better compared to others in the lab, including their work ethic, intellectual contributions, day-to-day lab maintenance, feedback, independence, communication of progress and challenges, and overall productivity (**Figure 4n–q**). Ratings of independence (**Figure 4r**) and communication (**Figure 4s**) trended higher for interns than their peers with 42% and 33%, respectively, being rated as higher or much higher than non-interns. On survey items about the impact of the program on interns' research, 92% of research advisors believed that the program had a neutral or positive impact on the *quantity* of research their trainees were able to accomplish (**Figure 4m**), and 96% reported a neutral or positive impact on the *quality* of research (**Figure 4l**). Overall, 85% of interns met or exceeded their research advisors' expectations in the lab (**Figure 4k**), and 89% of research advisors were somewhat to very likely to encourage a trainee to participate in the future (**Figure 4i**).

## Challenge of matching dissertation project to internship focus

Alumni interview data suggest that when internships strongly align with students' dissertation projects the benefit for interns increases. However, in some cases close alignment introduces tension due to the company's interest in protecting their intellectual property from being exposed prematurely by the intern. For example, one issue that surprised many research advisors was the host sites' requirement of confidentiality, which was particularly stringent for industry-based research and development internships. Some interns felt they could not discuss any aspects of the internship with their research advisor, because they were unclear what, if any information they could share, and this frustrated some research advisors. Because of this potential complication, our advice to interns is to choose an internship project that is tangential to their dissertation research. Furthermore, in our experience, the ideal internship project overlaps on a technical level, but not a scientific question. This allows the acquisition of new experience and skills without complicating the completion of degree-related research.

## Communication with the research advisor

Most research advisors indicated that they had received no communication from the hosts of their trainees at the internship sites, and they would have liked to have had some interaction. They reasoned that if they are the research advisor, they should receive some information on the internship training received. Research advisors emphasized that they did not want long reports or face-to-face meetings, but rather a summary of what had occurred during the internship. This is a simple change that we plan to implement going forward.

Most research advisors stated in focus groups that they had no recollection of any communication problems with the program leadership, so they concluded that it must have been sufficient. They did not consider communication with program leaders to be necessary or critical to the success of the internships. This finding is supported by quantitative crosstabs structured from research advisor surveys (*Figure 4u*). For example, the frequency of communication with program staff was considered 'sufficient' by 79% of surveyed faculty, even though in 53% of cases research advisors also indicated that they 'never', 'very rarely', or 'rarely' communicated with project leaders (*Figure 4u*). Twenty-one percent of research advisors indicated that there was not enough communication with project leaders. In nearly all cases where communication was rated as insufficient, communication was reported as being infrequent. These data indicate that effective communication between program leadership with

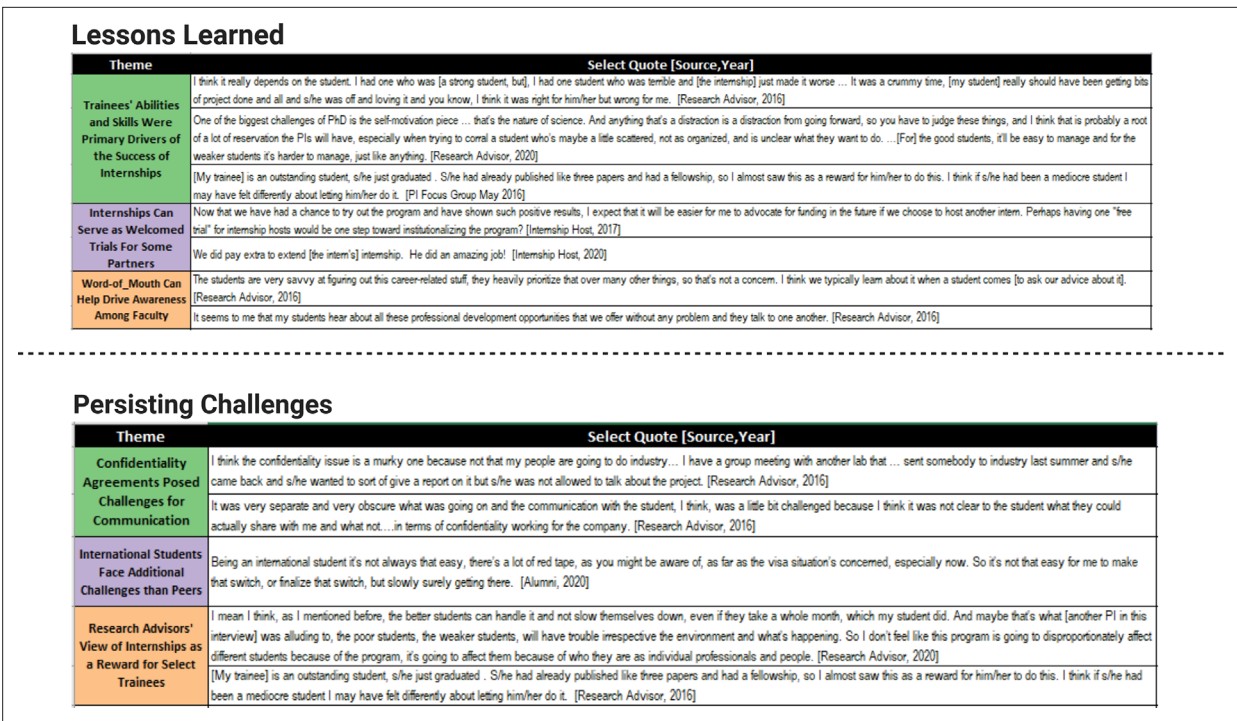

**Figure 5.** Lessons learned and persisting challenges (qualitative evidence).

research advisors is important for program buy-in, but that it does not need to be extensive. *Figure 5* contains quotes from interns and faculty advisors that represent key points related to lessons learned and persisting challenges to this internship program.

### Challenges for international trainees

During the life of the ImPACT program, federal immigration statutes and the university's interpretation of those guidelines have been dynamic. This has made it difficult, and sometimes impossible, for international trainees to participate in internships. An advantage of the funding model in which interns remain on institutional payroll during the internship is that it is easier for international students to be approved for some internships. We have also had limited success with creating a credit-bearing course that international students can enroll in during their internship; however, in that case, a tradeoff is that additional tuition credits must be paid for by the research advisor or the intern. Credit-bearing courses can make it easier for trainees on visas to use their allowance of Curricular Practical Training (CPT) during an internship. CPT is more freely available than Optional Curricular Training (OPT) allowances for most visa holders, and CPT can be approved by the university's international student office whereas OPT must be approved by the federal government. At times, the only option for an international trainee to participate in an internship has been to use OPT allowances, but this process is expensive and time-consuming to apply for and is usually not the preferred path. In addition, data from alumni interviews suggest that the movement into desired careers, irrespective of career sector, may be particularly difficult for international students, even if they had very productive internships. Overall, it is important to fully explore opportunities that may work for international trainee internships at each institution and make as many available as possible to customize the best-fit options for each trainee.

### Funding challenges and options

The funding for the ImPACT internship program – while coming from institutional sources – was made possible from 2015 to 2019 by savings in other areas due to UNC's NIH BEST award. Due to the success of the program during the grant period, the UNC School of Medicine has continued to fund the program since 2020. Regardless of the funding source, our model was designed to maximize faculty support by ensuring that research advisor grant funds were not used to support the trainee during the internship. Removing the burden of deciding how much time out of the lab is acceptable while paid on a research grant can make it easier for faculty to support a trainee's internship.

Since the inception of the internship program, we have experimented with multiple funding streams to provide sustainable internship opportunities. Some examples include large- and small-scale grants, industry-funded partnerships, institutional funding, and partnering with granting agencies that fund individual fellowships. The latter is worth discussing further. We have found that it is worthwhile to ask program officers of individual fellowships if a trainee funded by their organization can do a 160-hr internship as part of their training plan. Such fellowships include F31 and F32 NRSA fellowships from NIH, the National Science Foundation Graduate Research Fellowship, and the Howard Hughes Medical Institute Gilliam Fellowship. Each of these funding organizations has allowed some or all of their fellows who request it permission to remain on fellowship funding during their internship. While each funding model has its own pros and cons, we have found that securing university funding as the primary source of funds helps us to retain control of internship length, which, as discussed above, is critical to program sustainability and has helped our program to flourish. Relatedly, collecting program evaluation data over time has allowed program directors to make a case for the value of continued funding to support the internship program.

## Discussion

This mixed-methods study evaluating the benefits of UNC's internship program for doctoral-level trainees and postdocs demonstrated a wide range of positive results. Both qualitative and quantitative data pointed to definitive benefits for each of the stakeholder groups (trainees/interns, research advisors/faculty/labs, and internship hosts). Career outcomes were positively impacted by participating in the internship program. Finally, faculty attitudes show a significant shift toward supporting trainee participation in an internship program following the initial implementation.

## Summary of benefits to stakeholders

### Benefits to interns

Overall, interns found the internship experience highly valuable, with nearly all (95–97%) agreeing that the internship experience provided networking and professional development opportunities, expanded their network, and increased their competitiveness for the job market. The internship program was purposefully designed to provide career exploration and skill training, *not* job placement; nonetheless, positive career matches between internship field and first job placement were expected and achieved. Participation in the internship program increased the likelihood of matching one's career interest with a first job placement in the trainees' field of choice by threefold in comparison to non-participants.

Second, the program was explicitly designed to minimize any potential negative impacts on the trainees' research experience – this was achieved in that a large portion of interns even agreed that their research productivity and quality increased (42% and 33%, respectively; *Figure 4d and h*). Likewise, any negative impacts were minimized in that even for the minority (23% and 27%) that may have experienced temporary reductions in research productivity or quality, our empirical evidence suggests that no long-term deleterious effects were detected in publication rates for total or first-author publications or in terms of time to degree. This is in line with and expands upon our previous work showing no detriment to trainees' productivity and efficiency based on overall professional development participation across 10 institutions nationally (*Brandt et al., 2021*). The current data extend and replicate our initial findings in the aforementioned report that trainee productivity was not negatively impacted by participating in internships. The absence of a measurable cost to time or productivity suggests that internships provide a multitude of measurable benefits to multiple stakeholder groups and have a high cost-to-benefit payoff.

### Benefits to host companies

Hosts generally had positive expectations of accepting an intern, which was reinforced by their experiences. While hosts ranked mentoring opportunities as the top initial motivator for taking on an intern, they found better than expected experiences with intern assistance with products and gaining fresh ideas. Furthermore, while they expected a benefit from interacting with a potential future hire, this expectation was ranked even more positively than expected by the end of the hosting experience.

### Benefits to research advisors

As a stakeholder group, it is important to acknowledge that faculty research advisors take on the most risk and have the least to gain from encouraging their trainees to participate in internships outside of the lab. Yet, research advisors reported tangible benefits to the lab in both quantitative survey results and qualitative focus group responses.

Recruiting high-quality, career-motivated, students to graduate programs is an indirect benefit of having an internship program. Prospective and incoming graduate students consistently rate professional development offerings as a major factor in choosing UNC on our annual recruitment surveys (Dave McDonald, unpublished data). The increasing quality of incoming students, especially those who communicate their interest in internships from the beginning, is recognized by the faculty and is an important driver of faculty support for the program. Another ancillary benefit of experiential career exploration programs, including internships, is the advantage that such programs offer to a university in securing and maintaining institutional training grants such as T32 Institutional Research Training Grants from the NIH.

## Sustainability and faculty support

It is important for internship programs to lay the groundwork for a successful program by encouraging trainees to strategize their choice of internship timing to maximize gain and minimize impacts on their productivity. In addition, communicating with faculty before and after implementation is critical to gain widespread buy-in, manage expectations, and change course as needed. Even though the internship program began with relatively strong faculty support, that support increased substantially for both the 1-month (more popular) and the 3-month (less popular) internship formats as measured in a survey of active training faculty after the program was 5 years old. We believe this support is a result

of successfully balancing stakeholders' interests and concerns appropriately, especially keeping the length of the internship relatively short and ensuring that there is a minimal financial cost to research advisors.

There tends to be a concern among faculty that exploring other career options could dissuade trainees from pursuing a tenure-track faculty career. While this may be true for some trainees, evidence suggests that career exploration can lead other trainees who were initially *not* considering academic careers to in fact *seek out* academic opportunities once they understand their options in more detail (*Layton et al., 2020*). This is likely due to non-academic career paths holding inflated appeal due to the 'grass-is-greener-elsewhere' phenomenon. When trainees engage in experiential career exploration, they return with a more realistic understanding of workplace realities. Wherever such experiences direct a trainee, an informed career decision is in the best interest of the trainee and their future employer.

## Other programmatic considerations

One consideration when planning for an internship program is the level of resources and personnel that are needed to build and sustain the program. Staff time allocation and personnel effort were cited by some BEST programs as impediments to developing internship programs (*Lenzi et al., 2020*). At UNC, one PhD-level staff director is primarily responsible for the internship program, which takes about 40% of their full-time effort. One other PhD level director lends their time as needed (e.g., supporting exploration of niche careers/internships, assisting with placements to unusual internship formats, connecting interns to additional networks). Importantly, there is also an active faculty advisor and advocate who devotes 5–10 hr per month to support the program (e.g., strategic planning, program evaluation analysis, administrative oversight, connecting with faculty, and answering common questions). The most time-consuming parts of program administration are handled by the primary program director, and include managing and growing partnerships; communicating with interns, supervisors, and research advisors; coaching interns; overseeing the application and matching process; and coordinating financial funding source management.

During the first 5 years of the internship program, internships were available to postdoctoral trainees and graduate students. After 2020, our funding mandate no longer allowed us to fund postdoctoral internships even though a fraction of internship hosts have a stated preference for more senior postdoctoral trainees. When postdocs were included, roughly one in five (21%) of interns were postdocs. We found that research advisors of postdoc interns were less likely to be supportive of the program for several reasons. First, a 160 hr internship may represent a greater percentage of the typical postdoc employment period, given the variation of postdoctoral training length. Second, some advisors view the postdoctoral experience as less amenable to career exploration. Third, given the at-will employment designation of postdocs, they are more likely to end their training abruptly if offered a full-time position by the host company. Another challenge of funding postdoctoral internships is that a senior postdoc's salary can be close to twice that of a graduate student's stipend. Notwithstanding these challenges, postdoc interns rated their experience highly and often transitioned seamlessly into employment with their internship host.

## Ongoing challenge: Integrating off-campus skill acquisition into graduate and postdoctoral training

Without a doubt, most any trainee would benefit from an off-campus internship, even if they plan to remain in an academic career setting. Indeed, internships are ubiquitous in many other professional training programs, such as law, business, nursing, computer science, and engineering programs (*Van Wart et al., 2020*). However, due to the scarcity of resources and the current structure and incentives inherent to biological and biomedical graduate and postdoctoral training, the internship process is a highly competitive process accessible only to a fraction of trainees and at only a minority of academic institutions. The competitive process benefits internship hosts and program reputation, but it leads to other challenges. For example, some research advisors maintain the view that an internship is an experience that only the most productive students should benefit from. One research advisor in a focus group interview stated, 'Research advisors use the internship opportunity as a reward for outstanding students'. As the value of internships for life science PhD and postdoc trainees become more evident, and as graduate/postdoc training programs and science workforce

employers experience the benefits firsthand, we expect to see continued systemic changes that will increase access to internships and shift some of the costs away from training programs and universities. We hope that our experiences and lessons learned from the implementation of the internship program will help to encourage other institutions to either initiate or extend their biomedical internship possibilities.

Successful internship models at other universities that should be considered by universities developing their own programs include the University of California San Francisco (*Schnoes et al., 2018*), which places their interns after dissertation defense and PhD graduation. This removes potential conflicts of interest between the research advisor and student, but it may require that the intern be applying for full-time positions before the full benefit of the internship is realized. Other models include shorter internships, unpaid internships, and company-paid internships (e.g., Rutgers iJobs program, University of Rochester BEST internships, the ASPIRE program at Vanderbilt University, Cornell University BEST Internships). Future work should include evaluating whether the stakeholder benefits and faculty buy-in are similar or different from these and other internship models. The current model has the benefit of occurring in the midst of training, which allows follow-up skill-building and reflection before the trainee enters the job market. On the other hand, from the perspective of a company that views the internship as a hiring mechanism, the delay between the end of the internship and the start date of the trainee could be a downside of this model.

## Limitations

Limitations to the generalizability of our program and results include differing institutional climate and culture, the local economic environment, and the number and type of local employers. First, there may be funding structures and sustainability plans that are a better fit to universities of different sizes and compositions. We acknowledge the need for future studies to evaluate the feasibility and outcomes of internship programs funded via different models to see if faculty support and student outcomes would be comparable under different models. Second, not all universities may be able to create a local internship program, especially those not situated near a strong biotech or pharma hub. Third, geographic challenges can be overcome by alternate program designs, including virtual internships, that utilize partnerships with institutions or companies in other locations. Another alternative is traveling to biotech and pharma hubs such as Boston and San Francisco, either for short or longer experiential learning opportunities. Such programs have been piloted by the University of Chicago and Vanderbilt University (see *Van Wart et al., 2020*). Another alternative is to find internship partners in areas of strong economic opportunity in the local area, even if those may not be a direct match to the disciplinary skill sets. Transferrable skills of graduate students can be emphasized when exploring these partnerships (e.g., *Christine and Judith, 2020*).

A potential limitation inherent to this type of observational research is a self-selection bias that may affect those who choose to participate in internships (see *Brandt et al., 2021* for further discussion). It is possible that interns differ from non-participants in career goal clarity, motivation, research advisor support, or other undefined ways. Hence, a randomized-controlled trial in a scaled-up program that includes a variety of experiential learning options would shed light on which programmatic aspects or experiences are most beneficial. Trainees could be ethically randomized and assigned to one of a variety of career development and experiential learning opportunities to better empirically ascertain the best programmatic elements. However, as in clinical trials, once a treatment (or, in our example, a career development program such as internships) is shown to provide definitive benefits, it would be unethical to assign people to a true control condition that deprives them of that opportunity. In addition, future research should examine ways to create internship opportunities that are available for all students in a way that fully integrates an internship experience into the graduate training program.

Furthermore, a key component of experiential learning includes reflection to reinforce the benefits of experiential learning to trainees (*Van Wart et al., 2020*). While trainees had the experience to reflect on their internship while providing feedback during program evaluation, a more structured and in-depth guided reflection might provide additional benefits. Future programs and studies should systematically examine to what extent additional post-internship reflections and/or ongoing regular reflections during the internship experience may provide a richer gain in perspectives. Other future studies could probe faculty advisor support for internships at institutions beyond our own since training culture and faculty perspectives are influenced by many factors and vary from institution to institution.

### Adapting to national trends: Customizable program design

Global events, including the COVID-19 pandemic, added to the need for creative and innovative experiential career development opportunities amid an ever-changing academic job market (*Mathur, 2020*). Hence, during later cohorts, we needed to keep our model nimble to adapt to changing workforce needs, current job market trends, as well as limitations to in-person internship opportunities. As we emerged from the negative job market impacts during earlier phases of the pandemic, we are now entering a phase of increased hiring and the unexpectedly high level of opportunity for trainees now entering the job market (e.g., the Great Resignation, multiple job opportunities per person available nationally; *Gewin, 2022*) we consider future directions for the internship program model to expand into new territories and develop partnerships between institutes and organizations. This includes job location flexibility (e.g., remote, hybrid, in-person), adjusting company needs/interests to still meet the needs of trainee career interests, and multiple funding models that have worked to create sustainable programming.

### Conclusions

Internship programs for PhD and postdoctoral level life scientists provide a myriad of benefits to interns, the research advisors' lab climate and productivity, and the internship host organizations. A customized internship program that maximizes an institution's resources, location, and access to partner organizations can result in a sustainable internship program. Implementation of the program in a manner that aligns and balances the interests of all three principal stakeholders as well as secures research advisors' support is critical to the success of the program.

## Methods

### Overview

The methods used span stakeholder groups and research questions; hence, they are not presented as separate studies but rather referred to as relevant when each topic, theme, or relevant stakeholder group is discussed. An overview of each method of data collection is included here, with additional detail about planned analyses and results included within each subsequent section. Qualitative data collection included focus groups and open-ended survey response options, whereas quantitative data collection included Likert-type survey responses and career outcome data and matchings. Since these data were collected across multiple types of surveys and databases, the 'Methods' section details each data collection methodology for the reader to refer to.

### Stakeholder data elements

| Stakeholder | | Faculty attitude survey (Pre) | (Post) | Internship surveys (Pre) | (Post) | Focus group interviews (Year 1) | (Year 5) | NIH BEST entrance survey | Career outcomes census |
|---|---|---|---|---|---|---|---|---|---|
| | Current intern | | | x | x | x | | x | |
| | Alumni intern | | | | | | x | | |
| Trainees | All trainee alumni | | | | | | | | x |
| | Active training faculty | x | x | | | | | | |
| Faculty | Research advisors | | | x | x | x | | | |
| Industry | Internship hosts | | | x | | | | | |

### Faculty attitude survey (pre-/post-program implementation)

Identical surveys asking current active training faculty to provide their opinions on trainees participating in 1-month-long and 3-month-long internships were administered approximately 5 years apart (pre- and post-NIH BEST funding and the corresponding implementation of the internship program). Surveys were administered via Qualtrics before BEST award (2014, n = 112–114) and following-NIH BEST intervention (2019, n = 117–118), with comparable response rates each iteration. Emailed survey

links requested voluntary completion of the survey via the biomedical sciences umbrella program list-serv which reaches roughly 300 active training faculty across 14 departments.

### Internship surveys (pre-/post-survey data collection)

For program evaluation, responses were requested from all participating interns, hosts, and research advisors via personalized email invitation. Data was collected via SurveyMonkey by the external evaluator to ensure confidentiality and candor. Two reminders were sent to urge non-completers to respond.

|  | Research advisors | Internship hosts | Current interns (pre) | Current interns (post) |
|---|---|---|---|---|
| Cohort 1 | 12 | 15 | – | 13 |
| Cohort 2 | 14 | 19 | 23 | 16 |
| Cohort 3 | 21 | 22 | 27 | 24 |
| Cohort 4 | 15 | 17 | 13 | 19 |
| Cohort 5 | 13 | 17 | 20 | 8 |
| Total | 75 | 90 | 83 | 80 |

## Focus group interviews

### Faculty (Year 1 and Year 5)

A subset of seven research advisors who were mentoring trainees in the program during the 2015–2016 academic year were interviewed in a focus group format in March 2016. Interviewees included mentors of both postdocs and graduate students, some of whom had completed their internships and some of whom were still in the planning stages, as well as one faculty member who had not yet mentored an intern. A follow-up focus group session was conducted in June 2020, with five of the seven original research advisors available to offer reflective thoughts on the program.

### Intern (Year 1 and Year 5)

A subset of 10 trainees in the program during the 2015–2016 academic year were interviewed in a focus group format in March 2016. Interviewees included both postdocs and graduate students, some of whom had completed their internships and some of whom were still in the planning stages. Follow-up focus group interviews were conducted with a random sample of program alumni who had completed internships in the 2015–2016 academic year. A total of eight alumni of the program were interviewed across two focus groups sessions in September 2020.

Interviews conducted during the 2015–2016 academic year were in person, while all interviews in 2020 were virtual through the Zoom platform. During the interviews, no project leaders or university administrators were present for either format, and each focus group session lasted roughly an hour. All sessions were audio-recorded, transcribed, and analyzed for themes. We used ATLAS.ti version 6, a computer-assisted qualitative data analysis software package (GmbH, Germany, 1993–2020). Two members of the external evaluation team (K. Wood and D. Whittington) developed the list of themes and prepared summaries without identifiers for project leaders.

### NIH BEST entrance survey

Career interests for controls and interns were gathered from NIH entrance survey sent to all UNC life science graduate students in 2015 included 342 students who initiated the survey and 301 who completed it; for postdocs, 332 initiated and 273 completed the survey. For any interns who did not complete the entrance survey (such as those who entered UNC after 2015), a proxy interest match was determined using a pre-internship career interest rating submitted as part of the internship application (n = 69 of 116 total interns in the matching analysis).

### Career outcomes census

Career outcomes were collected for all graduate student alumni on the bi-annual census, along with all postdocs who either completed an entrance questionnaire or pre-internship questionnaire

indicating career interests. Career outcomes were collected using publicly available information found on LinkedIn, laboratory websites, personal websites, company websites, social media (e.g., Twitter), PubMed, Google Scholar, etc. Whenever possible, two sources of information were used to corroborate the current job title and employer.

## Participants

Survey data was requested from all interns, their respective faculty advisors, and internship hosts. Faculty attitude surveys invited all active training faculty to participate (defined as all faculty who have supervised graduate students in the previous 5-year period). Focus groups included interns, alumni, and faculty invited to participate to provide program feedback.

## Career interests and first position matching analysis

Career outcomes matches were evaluated for those who had exited the program to attain a first-position title (all interns and non-interns from the BBSP program are included in our career outcomes database from our bi-annual census). For any trainee who stayed in their graduate lab as a postdoctoral trainee for less than 12 months, their first position outside the organization was instead defined as the first position. This sample included graduates and postdoctoral scholars who transitioned to their first career position between May 2014 and August 2021.

Career matches were calculated by the match between career interests with career outcomes taxonomy. Career interest measures were based on a 5-point Likert scale (Not at All Considering, Slightly Considering, Moderately Considering, Strongly Considering, Will Definitely Pursue). For each survey respondent, we identified their top choice(s) by determining one or more career path(s) that they ranked most highly (e.g., the one that they reported they would 'definitely pursue'). If a less career-confident respondent did not rate any career path as 'will definitely pursue', then we used their next highest response(s). The total number of career interests rated as either 4 or 5 was compiled (the sum of the 20 career interest variables, where 1 indicated an interest of 4 or 5 selected, and 0 indicated any lesser selection of interest).

The 20 career interests reported on the NIH BEST baseline survey were indexed to align with the 24 bins in the Job Function tier of the 2017 Unified Career Outcomes Taxonomy (UCOT Exp2, Stayart et al). See OSF File 3: 'First position logistic regression data.xlsx' for a crosswalk between the 20 entrance survey options and the 24 job functions from NIH BEST career taxonomy. Group matches were defined by sorting the 20 or 24 categories respectively into eight umbrella career groups (see OSF File 3: 'First position logistic regression data.xlsx').

## Publication analysis

PubKeeper (Strategic Evaluations, Inc) was used to submit trainee and research advisor name pairings for automated publication hit queries on NCBI's PubMed database. PubKeeper returns PubMed IDs, PubMed Central IDs, author lists, and citation details including whether the trainee is a first author. This data was used to count the number of first-author publications and total publications for each trainee. Any trainee who had more than nine papers or less than one was analyzed by a member of the team to be sure that the number of publications was accurate. For any publication where the trainee was a second author, the PDF was accessed and examined for evidence that the trainee was a co-first author. If so, the publication was counted as a first-author paper. See OSF File 7: 'PubKeeper publication output plus analysis 2-14-2023.xlsx' for publication data.

## Data analysis

All survey data were stored and analyzed using SPSS (IBM, New York, NY). In the cases where survey data were collected at one-timepoint (e.g., internship hosts, exit-only items for current interns), the team computed descriptive statistics. Survey data collected across two-timepoints were tested for statistical significance using *t*-tests. To maximize sample size due to partial responses, independent samples *t*-tests were used for the pre-and post-survey data collected from the interns. Independent samples *t*-tests were used for the Faculty Attitude Survey data set, given those identifiers were not collected to allow a pairing of responses across the two-timepoints. Independent samples *t*-tests were also used for trainee publication to compare participant (interns) versus non-participant graduate

students on productivity (first author and total publications), including any program alumni who entered during the time range that the internship was offered (e.g., same start date as any participating interns in our sample). When testing for statistical significance, alpha criterion used was 0.05 (level of significance indicated for p-values<0.05, 0.01, or 0.001).

Qualitative data, for example, open-ended questionnaire responses and interview transcripts, were stored and analyzed via Atlas.ti (Berlin, Germany). The external evaluation team led the analysis of these data and incorporated an inductive approach, coding narrative segments within the raw data sets and searching for dominant and significant themes among the codes. The evaluation team then reported the most common themes to program leaders, including representative quotes that were removed from identifying information. A subset of these common themes, along with representative quotes, are included in this article.

Lastly, a binary logistic regression was used to examine the extent to which internship participation was related to successful career matches with expressed career interests (e.g., career interests indicated on baseline or pre-internship surveys associated with the first job in the field of interest). For the logistic regression, the team used NIH BEST entrance survey responses for all trainees, with any missing responses supplemented by pre-internship application data for postdocs and grad student interns who did not participate in the BEST entrance survey. The logistic regression model included participation in the internship program (0/1) and graduate student or postdoc status (0/1), while controlling for the number of total career interests (0–24). We felt it important to control for career interests since those who expressed more interests were mathematically more likely to find a match, and we hypothesized that the stage of career (grad student versus postdoc) was also important to assess due to the potential impact. A posthoc model also included social identities in the same model (gender, race/ethnicity, and citizenship); results from the initial model were consistent and hence the original model was retained (see *Tables 1 and 3*; data available in OSF File 1: 'Career Interests and Outcomes logistic regression.sav').

## Acknowledgements

The authors would like to thank Patricia Labosky for her continual support of the NIH Broadening Experiences in Scientific Training (BEST) institutions, their investigators, her leadership of the larger national initiative, and especially for encouraging us to disseminate the research about our ImPACT internship program. Funding sources include NIH BEST Award DP7OD020317 and NIGMS – Science of Science Policy Approach to Analyzing and Innovating the Biomedical Research Enterprise (SCISIPBIO) Award 1R01GM140282.

## Additional information

### Competing interests

Dawayne Whittington: Employee of Strategic Evaluation, Inc. The other authors declare that no competing interests exist.

### Funding

| Funder | Grant reference number | Author |
| --- | --- | --- |
| National Institute of General Medical Sciences | 1R01GM140282 | Patrick Brandt Christiann H Gaines Patrick Brennwald Rebekah L Layton |
| NIH Office of the Director | BEST Award,DP7OD020317 | Patrick Brandt Patrick Brennwald Rebekah L Layton |

The funders had no role in study design, data collection and interpretation, or the decision to submit the work for publication.

## Author contributions
Patrick Brandt, Rebekah L Layton, Conceptualization, Resources, Data curation, Formal analysis, Supervision, Funding acquisition, Validation, Investigation, Visualization, Methodology, Writing – original draft, Project administration, Writing – review and editing; Dawayne Whittington, Conceptualization, Data curation, Software, Formal analysis, Validation, Investigation, Visualization, Methodology, Writing – original draft, Writing – review and editing; Kimberley D Wood, Conceptualization, Data curation, Formal analysis, Investigation, Methodology, Writing – original draft, Writing – review and editing; Christopher Holmquist, Data curation, Formal analysis, Writing – review and editing; Ana T Nogueira, Data curation, Formal analysis, Visualization, Writing – review and editing; Christiann H Gaines, Resources, Formal analysis, Funding acquisition, Writing – review and editing; Patrick Brennwald, Conceptualization, Resources, Data curation, Formal analysis, Supervision, Funding acquisition, Validation, Investigation, Visualization, Methodology, Writing – original draft, Writing – review and editing

## Author ORCIDs
Patrick Brandt ![ORCID] https://orcid.org/0000-0002-6189-9051
Patrick Brennwald ![ORCID] https://orcid.org/0000-0001-9344-285X
Rebekah L Layton ![ORCID] https://orcid.org/0000-0001-7113-1348

## Ethics
All study activities were reviewed by the UNC Institutional Review Board (IRB# 14-0544) and conducted in accordance with ethical practices, including de-identifying data (see OSF data availability statement). Information sheets and additional consent information were provided as relevant to each respective participant population (e.g., focus group verbal consent; embedded survey consent information) in accordance with IRB-reviewed data collection procedures.

Reviewer #2 (Public review): https://doi.org/10.7554/eLife.91011.3.sa1
Author response https://doi.org/10.7554/eLife.91011.3.sa2

# Additional files

## Supplementary files
MDAR checklist

## Data availability
All original data that form the basis for this paper is available on the Open Science Framework website using the link: https://doi.org/10.17605/OSF.IO/ED2PG. Description of file names available on Open Science Framework (OSF): OSF File 1: "Career Interests and Outcomes logistic regression.sav", OSF File 2: "Faculty Attitude Survey.sav", OSF File 3: "First position logistic regression data.xlsx", OSF File 4: "Intern Data All Years (Pre-& Post-) - De-identified - July 7 2023.sav", OSF File 5: "Internship Supervisor Data All Years (Post-only) - De-identified - July 7 2023.sav", OSF File 6: "PI Data All Years (Post-Only) - De-identified - July 7 2023.sav", OSF File 7: "PubKeeper publication output plus analysis 2-14-2023.xlsx", OSF File 8: "publication outcomes.xlsx", OSF File 9: "TTD and other trainee details from database v39.xlsx".

The following dataset was generated:

| Author(s) | Year | Dataset title | Dataset URL | Database and Identifier |
|---|---|---|---|---|
| Brandt PD, Whittington D, Wood KD, Holmquist C, Nogueira AT, Gaines CH, Brennwald PJ, Layton RL | 2025 | Development and Assessment of a Sustainable PhD Internship Program Supporting Diverse Biomedical Career Outcomes | https://doi.org/10.17605/OSF.IO/ED2PG | Open Science Framework, 10.17605/OSF.IO/ED2PG |

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
