## [Editor Report · eLife assessment]

This **important** study evaluates the outcomes of a single-institution pilot program designed to provide graduate students and postdoctoral fellows with internship opportunities in areas representing diverse career paths in the life sciences. The data **convincingly** show the benefit of internships to students and postdocs, their research advisors, and potential employers, without adverse impacts on scientific productivity. This work will be of interest to multiple stakeholders in graduate and postgraduate life sciences education and should stimulate further research into how such programs can best be broadly implemented.

---

## [Referee Report · Reviewer #2 (Public review)]

Summary:

The authors describe five year outcomes of an internship program for graduate students and postdoctoral fellows at their institution spurred by pilot funding from an NIH BEST grant. They hypothesized that such a program would be beneficial to interns, internship hosts, and research advisors. The mixed methods study used surveys and focus groups to gather qualitative and quantitative data from the stakeholder groups, and the authors acknowledge that limitation that the study subjects were self-selected and also had research advisors who agreed to allow them to participate. Thus the generally favorable outcomes may not be applicable to students such as those who are struggling in the lab and/or lack career focus or supportive research advisors. Nonetheless, the overall finding support the hypothesis and also suggest additional benefits, including in some cases positive impact for the lab, improved communication between the intern and their research advisor, and an advantage for recruitment of students to the institution. The data refute one of the principle concerns of research advisors: that by taking students out of the lab, internships reduce individual and overall lab productivity. Students who did internships were significantly less likely to pursue postdoctoral fellowships before entering the biomedical workforce and were more likely to have science-related careers versus research careers than control students who did not do internships, although the study design cannot determine whether this is a causal relationship.

Strengths:

(1) Sample size is good (123 internships).

(2) Response rate is high, minimizing potential bias.

(3) The internship program is well described. Outcomes are clearly defined.

(4) Methods and statistical analyses appear to be appropriate (although I am not expert in mixed methods).

(5) "Take-home" lessons for institutions considering implementing internship programs are clearly stated.

Appraisal:

Overall the authors achieve their aims of describing outcomes of an internship program for graduate career development and offering lessons learned for other institutions seeking to create their own internship programs.

Impact:

The paper will be very useful for other institutions to dispel some of the concerns of research advisers about internships for PhD students (although not necessarily for postdoctoral fellows). In the long run, wider adoption of internships as part of PhD training will depend not only on faculty buy-in but also on availability of resources and changes to the graduate school funding model so that such programs are not viewed as another "unfunded mandate" in graduate education. Perhaps industry will be motivated to support internships by the positive outcomes for hosts reported in this paper. Additionally, NIH could allow a certain amount of F, T, or even RPG funds to be used to support internships for purposes of career development.

---

## [Author Response]

The following is the authors’ response to the original reviews.

**eLife assessment**
This important study evaluates the outcomes of a single-institution pilot program designed to provide graduate students and postdoctoral fellows with internship opportunities in areas representing diverse career paths in the life sciences. The data convincingly show the benefit of internships to students and postdocs, their research advisors, and potential employers, without adverse impacts on scientific productivity. This work will be of interest to multiple stakeholders in graduate and postgraduate life sciences education and should stimulate further research into how such programs can best be broadly implemented.

Thank you for your assessment. We agree that sharing our process for creating this internship program with the wider higher education community is important and we hope it will spur establishment of new programs at other institutions.

**Public Reviews:**

**Reviewer #1 (Public Review):**
The goal of this study was to determine whether short (1 month) internships for biomedical science trainees (mostly graduate students but some post-docs) were beneficial for the trainees, their mentors, and internship hosts. Over a 5 year period, the outcomes of trainees who completed internships were compared with peers who did not. Both quantitative results in terms of survey responses and qualitative results obtained from discussion groups were provided. Overall, the data suggest that internships aid graduate students in multiple ways and do not harm progress on dissertation projects. 'Buy-in' from mentors and prospective mentors appeared to increase over time, and hosts also gained from the contributions of the interns even in a short time period. While the program also appeared valuable for post-doctoral trainees, it was less favorably considered by post-doc mentors.

Thank you for such a positive and concise overview of this paper.

Strengths:The internship program that was examined here appears to have been very well designed in terms of availability to students, range of internship offerings, length of time away from PhD lab, and assessments.Having a built-in peer control group of graduate students who did not do internships was valuable for much of the quantitative analyses. However, as the authors acknowledge, those who did opt for internships are a self-selected group who may have character traits that would help them overcome the potential negative impacts of the internship.The quantitative data is convincing and addresses important considerations for all stakeholders.The manuscript is well-constructed to individually address the impact of the program on each set of stakeholders, while also showcasing areas of mutual benefit.The discussion of challenges and limitations, from the perspectives of participating stakeholders, program leaders, and also institutions, is comprehensive and very thoughtful.

Thank you for noting these strengths in experimental design, control group, and manuscript format.

Weaknesses:The qualitative data that resulted from the 'focus groups' of faculty mentors was somewhat difficult to evaluate given the very limited number of participants (n=7).

Thank you for pointing out the potential limitations of a small sample size. One reason we selected a qualitative approach to focus group data analysis in our experimental design was to supplement our larger quantitative analyses with faculty advisors. A benefit of relying on qualitative methods is that saturation of a representative set of themes can be reached even with a limited number of participants. This is particularly true when a homogenous sample is used, such as faculty members in the biomedical sciences (Guest, et al. 2006). We have added the following sentences at line 188 in the text to expand on the faculty focus groups:

“A group of faculty advisors in a range of disciplines and demographics, all of whom were active mentors with extensive training experience were invited to participate in the focus groups. Seven faculty advisors participated in the Year 1 focus group and 5 of those same 7 participated in Year 5. Saturation can occur with as little as six interviews in homogeneous samples (Guest et al. 2006) such as our biomedical faculty research advisors at a single institution.”

In the original analysis, we increased the generalizability of our findings by gathering faculty opinions and feedback using multiple methods. For example, faculty post internship surveys responses were returned by 75 faculty members over a 5-year period, which represents a 61% response rate. (Faculty post internship surveys results are shown in Figure 1, panels v-x and Figure 4, panels i-t.) In addition, the survey gauging general faculty advisor support for the program (Figure 3); which was administered two times, 4 years apart; gathers the opinions of 115 advisors in year 1 and 122 advisors in year 4. Thus, the faculty focus group surveys were only one of 3 ways that faculty input was gathered. In sum, while the small number of faculty mentors who participated in the focus groups has the potential to introduce bias, we made a conscious decision to use a mixed methods approach to expand beyond one sample to increase the generalizability of our results. However, to acknowledge the complexity of faculty advisor views on internships, we have noted the need to further study faculty advisor support for internships in broader samples as a future direction. This is the new wording we included at line 788:

“Other future studies could probe faculty advisor support for internships at institutions beyond our own since training culture and faculty perspectives are influenced by many factors and vary from institution to institution.”

Overall, the data support the authors' conclusions with respect to the utility of internship programs for all stakeholders. As the authors note, the data relate to a specific program where internship length was defined, costs were covered by a grant or institutional funding, and there were multiple off-site internship hosts available. Thus, the results here may not replicate for other programs with different criteria.

Thank you for noting these advantages that contributed to the success of this program. We agree that other institutions will encounter unique challenges when implementing their own internship program and have addressed some of these limitations in our discussion section. In the Discussion section of the paper, we outline considerations and review lessons learned in an effort to help others know what aspects of the program might or might not work in distinct situations or locations. We also point the reader to distinct internship models at other institutions in the hope that any university hoping to provide their trainees with internship opportunities can benefit from the collective experience of the relatively few programs that have found sustainable ways to accomplish this.

This work provides a valuable assessment of how relatively short internships can impact graduate students, both in terms of their graduate tenure and in their decision-making for careers post-graduation. As more graduate programs are heeding calls from funding agencies and professional societies to increase knowledge about, and familiarity with, multiple career paths beyond academia for PhD students, there is a need to evaluate the best ways to accomplish that goal. Hands-on internships are valuable across many spheres so it makes sense that they would be for life science graduates too. However, the fear that time-to-degree and/or productivity would be negatively impacted is important to acknowledge. By providing clear data that this is not the case, these investigators have increased the likelihood that internships could be considered by more institutions. The one big drawback, and one that the authors discuss at some length, is the funding model that could enable internship programs to be used more widely.

Thank you for providing suggestions to improve the generalizability of our results. We agree that finding a sustainable source of funding for internship programs, and the staff who direct them, is a primary obstacle to implementing these programs more widely. We provide some ideas and funding models for other institutions to consider, and future directions could examine internships that are un-funded or funded primarily by fellowships from supportive granting agencies. Accordingly, we have added the following text to future directions at Line 755:

“We acknowledge the need for future studies to evaluate the feasibility and outcomes of internship programs funded via different models to see if faculty support and student outcomes would be comparable under different models.”

**Reviewer #2 (Public Review):**
Summary:The authors describe five-year outcomes of an internship program for graduate students and postdoctoral fellows at their institution spurred by pilot funding from an NIH BEST grant. They hypothesized that such a program would be beneficial to interns, internship hosts, and research advisors. The mixed methods study used surveys and focus groups to gather qualitative and quantitative data from the stakeholder groups, and the authors acknowledge the limitation that the study subjects were self-selected and also had research advisors who agreed to allow them to participate. Thus the generally favorable outcomes may not be applicable to students such as those who are struggling in the lab and/or lack career focus or supportive research advisors. Nonetheless, the overall findings support the hypothesis and also suggest additional benefits, including in some cases positive impact for the lab, improved communication between the intern and their research advisor, and an advantage for recruitment of students to the institution. The data refute one of the principal concerns of research advisors: that by taking students out of the lab, internships reduce individual and overall lab productivity. Students who did internships were significantly less likely to pursue postdoctoral fellowships before entering the biomedical workforce and were more likely to have science-related careers versus research careers than control students who did not do internships, although the study design cannot determine whether this was due to selection bias or to the internship.

Thank you for such a positive and concise overview of this paper.

Strengths:(1) The sample size is good (123 internships).(2) The internship program is well described. Outcomes are clearly defined.(3) Methods and statistical analyses appear to be appropriate (although I am not an expert in mixed methods).(4) "Take-home" lessons for institutions considering implementing internship programs are clearly stated.

Thank you for enumerating these strengths. We also hope that the sample size, positive outcomes, and take-home lessons will be of benefit to other institutions.

Weaknesses:(1) It is possible that interns, hosts, and research advisers with positive experiences were more likely to respond to surveys than those with negative experiences. The response rate and potential bias in responses should be discussed in the Results, not just given in a table legend in Methods.

Thank you for noting this oversight. We were pleased that throughout our study, the majority of interns, faculty advisors and internship hosts responded to the surveys. As suggested, we have included the following text at line 132 in the first paragraph of the results section:

“The response rate for the 123 survey invitations sent to interns and their current research advisors and internship hosts ranged from 61% for research advisors to 73% for hosts, and about 66% for interns (averaging pre and post survey responses). In addition to quantitative surveys, qualitative themes and exemplars were collected from focus groups.”

(2) With regard to the biased selection of participants, do the authors know how many subjects requested but were not permitted to do internships?

We too were concerned about trainees who would not be able to secure their PI’s support to participate in an internship. Accordingly, as part of our program design and evaluation, in the inaugural year of the program our external evaluator, Strategic Evaluations, Inc, administered a survey to graduate students and postdocs who registered for an internship information session or who started, but did not complete the application. Registrants were asked about their decision to complete an application, their experience completing the application if they chose to do so, and the likelihood that they would apply to the program next year. Of the respondents, only 9% indicated that lack of PI support prevented them from participating (n=53 respondents). Hence while we cannot completely rule out PI support as a barrier, only a small percentage of trainees reported this as a barrier despite a robust response rate (43%). A second line of evidence that there was not a large number of students who were prevented from doing an internship by their research advisor is the high faculty approval rating of the program which was gathered in both year 1 and year 4 of the program (see figure 3). These two independent lines of evidence diminish our concern that faculty advisor resistance was a significant barrier to participation.

(3) While the authors mention internships in professional degree programs in fields such as law and business, some mention of internship practices in non-biomedical STEM PhD programs such as engineering or computer science would be helpful. Is biomedical science rediscovering lessons learned when it comes to internships?

Excellent point. We noted that internships are common in non-biomedical STEM masters and PhD programs, but we did not list experiential rotations and internships that are common in nursing, engineering, computer science and other such programs. We agree that many lessons learned from internships in all fields are transferable to the biomedical fields, and we also strongly believe that findings there need to be replicated in the biomedical sciences because of the unique funding model, incentive structure, and apprentice structure of the biomedical training. In response to this critique, we added the following text to the manuscript at line 724:

“Internships are ubiquitous in many other professional training programs such as law, business, nursing, computer science, and engineering programs (Van Wart, O’Brien et al, 2020).”

(4) Figure 1 k, l - internships did not appear to change career goals, but are the 76% who agreed pre-internship the same individuals as the 75% who agreed post-internship? What percentage gave discordant responses?

While our data cannot directly address this question as collected, we surmise that because internships in this program usually occur in the final 12-18 months of training and because there is an emphasis on the internship being a skill-building and not necessarily a career exploration initiative, therefore we were not surprised to see that the internship doesn’t radically alter many trainees’ career plans. One limitation of our study is that career goals were defined by pre-surveys at different timepoints depending on what stage of training an individual (whether control or internship participant) happened to be at during the administration of the baseline survey. We know from previous work that career goals often shift during training (see Roach and Sauermann, 2017 PLOS One, https://doi.org/10.1371/journal.pone.0184130, and Gibbs et al, 2014, PLOS One, https://doi.org/10.1371/journal.pone.0114736), so the point at which career interests are gathered makes a difference in this kind of analysis. Hence, we have expanded our discussion of this limitation to better acknowledge this critique beginning at Line 319.

“Because of the variable timing between pre-internship career interest surveys among interns and control trainees and securing the first job, future studies could more rigorously evaluate changes in career preferences between pre and post internship with an analysis that considers the time that has elapsed between career interest noted pre-internship vs post internship career placement. “

Appraisal:Overall the authors achieve their aims of describing outcomes of an internship program for graduate career development and offering lessons learned for other institutions seeking to create their own internship programs.

We thank you for your thorough reading and review of the manuscript.

Impact:The paper will be very useful for other institutions to dispel some of the concerns of research advisers about internships for PhD students (although not necessarily for postdoctoral fellows). In the long run, wider adoption of internships as part of PhD training will depend not only on faculty buy-in but also on the availability of resources and changes to the graduate school funding model so that such programs are not viewed as another "unfunded mandate" in graduate education. Perhaps the industry will be motivated to support internships by the positive outcomes for hosts reported in this paper. Additionally, NIH could allow a certain amount of F, T, or even RPG funds to be used to support internships for purposes of career development.

Thank you. We share your hope that the information and data resulting from this study will be valuable to other institutions. Your point about NIH (and other funders, for that matter) allowing trainees to participate in internship experiences while funded by the granting agency is an excellent one. We have found that communication with program officers often garners their support for the intern remaining on a fellowship or training grant during the internship. This allows the internship program to fund additional interns, especially those that are supported by the faculty advisor’s grants.

**Recommendations for the authors:**

**Reviewer #1 (Recommendations For The Authors):**
Two minor points about the comments used from focus groups.(i) In figure 5, there is a specific quote about being a reward that is used twice;(ii) It seems that there should be some consistency in how these quotes are relayed with respect to gender identification of the trainee. In some cases 's/he' is used, in others 'he' or 'she' is used, and in others 'they' is used.

We appreciate this suggestion and agree that a non-gendered convention would clearer – accordingly, we have revised all quotes to use “they” to be more consistent. In addition, we have removed the duplicated quote from figure 5, which was originally inserted in two sections because of its applicability to both the “Persisting Challenges” and “Trainees’ abilities and skills were primary drivers of the success of the internship”.

**Reviewer #2 (Recommendations For The Authors):**
(1) The paper is somewhat lengthy. Some redundant material can be eliminated - Lines 366-371 simply restate the data in Table 5. Lines 393-396 restate the data in Figure 3. The text should be reserved for interpreting rather than restating the data in tables and figures.

Thank you for this feedback and we agree that these sections can be condensed. We have removed some of the redundancy and retained enough for figures and text to each be stand alone for accessibility to the readers.